# Advances in Epstein–Barr Virus Detection: From Traditional Methods to Modern Technologies

**DOI:** 10.3390/v17081026

**Published:** 2025-07-22

**Authors:** Yidan Sun, Shuyu Ling, Dani Tang, Meimei Yang, Chao Shen

**Affiliations:** 1China Center for Type Culture Collection, Wuhan University, Wuhan 430072, China; yidan97@whu.edu.cn; 2College of Life Sciences, Wuhan University, Wuhan 430072, China; lingshuyu@whu.edu.cn (S.L.); danietang@whu.edu.cn (D.T.)

**Keywords:** Epstein–Barr virus (EBV), detection methods, point-of-care testing (POCT)

## Abstract

The Epstein–Barr virus (EBV) is a prevalent virus linked to various diseases, including infectious mononucleosis (IM), nasopharyngeal carcinoma, and Hodgkin’s lymphoma. Over the past few decades, EBV diagnostic strategies have evolved significantly—progressing from traditional serological assays and histopathology to more sensitive and specific molecular techniques such as nucleic acid amplification and high-throughput sequencing (HTS). While conventional methods remain valuable for their accessibility and established clinical use, they are often limited by sensitivity, speed, and multiplexing capability. In contrast, emerging technologies, including isothermal amplification, Clustered Regularly Interspaced Short Palindromic Repeats (CRISPR)-based diagnostics, multi-omics integration, and AI-assisted analysis, have demonstrated great promise in improving diagnostic accuracy, speed, and applicability in diverse clinical settings, including point-of-care testing (POCT). This review systematically explores the historical development of EBV diagnostic technologies, highlighting key milestones and future trends in precision medicine and global health readiness.

## 1. Introduction

The Epstein–Barr virus (EBV), a ubiquitous human γ-herpesvirus, is linked to various malignant and non-malignant conditions [1]. With over 95% of the adult population worldwide infected, its high prevalence and ability to remain latent while potentially causing cancer underscore the critical importance of early and precise EBV detection in clinical and epidemiological settings. Since its identification in 1964 [2], EBV detection methods have evolved significantly, driven by advancements in molecular biology, immunology, and medical diagnostics. These methods have transitioned from traditional microscopy and serological tests to more sensitive techniques such as nucleic acid amplification and high-throughput sequencing (HTS).

The outbreak of the Coronavirus disease 2019 (COVID-19) pandemic in late 2019 further accelerated advancements in viral detection technologies. The global demand for rapid, precise, and decentralized testing led to researchers creating and implementing innovative platforms, including isothermal amplification, CRISPR-based diagnostics, AI-assisted analysis, and portable point-of-care testing (POCT) devices. These innovations have not only transformed SARS-CoV-2 detection but also significantly improved EBV diagnostics, enhancing sensitivity, specificity, and speed.

In this context, a comprehensive review of the history of and recent developments in EBV detection methods is essential. This study systematically explores the evolution of EBV diagnostic technologies from the late 20th century to the post-COVID-19 era, emphasizing the transition from qualitative assays to smart, high-throughput, and portable solutions. By examining these advancements, this study will identify key milestones, assess the impact of the pandemic on EBV testing, and project future trends in precision medicine and global health readiness.

## 2. EBV Virology

### 2.1. Overview

The Epstein–Barr virus (EBV) is a member of the human herpesvirus (HHV) family, which consists of nine viruses, namely herpes simplex virus 1 (HSV-1), HSV-2, human cytomegalovirus (HCMV), varicella zoster virus (VZV), EBV, and human herpesviruses (HHV) 6A, 6B, 7, and 8 [3], distributed in three subfamilies (α, β, and γ). The EBV, also known as human herpesvirus type IV (HHV-4), belongs to the γ-herpesvirus subfamily known as the Lympho-cryptovirus genus [1]. It was first discovered in 1964 in tissue samples from children with Burkitt lymphoma [2]. The virus particles are spherical and consist of an envelope, a viral tegument, and a nucleocapsid extending from the outside to the inside. The nucleocapsid is an icosahedron formed by multiple capsid proteins surrounding a 172 kbp double-stranded DNA genome. The EBV carries more than 100 genes encoding about 85 proteins and about 50 non-coding RNAs [4].

The prevalence of EBV infection is exceptionally high, affecting over 95% of the adult population worldwide [5]. The EBV is recognized as an oncogenic virus that typically establishes latent infections after primary infection. As early as the late 1990s, the EBV was declared a class I carcinogen by the International Agency for Research on Cancer and the World Health Organization. Despite its pathogenic potential, the EBV has also contributed significantly to scientific research, particularly in the study of lymphocytes. For instance, EBV can infect B lymphocytes and convert them into continuously proliferating lymphoblastoid cell lines [6], making it a valuable tool for establishing immortalized B lymphocytes in human genetic studies.

The long-term latency of the EBV in lymphocytes can interfere with the body’s immune function and induce cellular proliferation and transformation. EBV infection involves multiple organs and systems, making it difficult to diagnose accurately. Therefore, early diagnosis and rational treatment are crucial for managing EBV-related conditions.

### 2.2. The Latent and Lytic Cycles

The EBV is primarily transmitted through saliva and targets lymphocytes and oropharyngeal epithelial cells. Following initial infection, the viral load of EBV DNA in saliva increases significantly and can persist for several months. Transmission can occur through activities such as kissing, sexual intercourse, hematopoietic cell transplantation, and solid organ transplantation. Studies have identified the periodic shedding of the EBV into the oral secretions of carriers [7]. The virus enters cells via endocytosis or fusion with the plasma membrane. Once inside the cell, the viral genome is delivered to the nucleus, where the EBV can either enter a latent phase or initiate lytic replication (Figure 1), depending on the cell type and microenvironment [8]. It is worth noting that during the latent phase, the virus is present in the nucleus in the form of circular episomes linked to the host genome chromatin using a viral protein called Epstein–Barr nuclear antigen 1 (EBNA-1).

Following initial infection, the EBV in B-cells typically enters a latent phase characterized by minimal viral gene expression, allowing for persistent infection in a dormant state. There are several types of EBV latency. Latency I represents a minimal viral activity state, enabling long-term persistence with limited immune detection. Latency II involves moderate viral activity and is commonly associated with epithelial and certain lymphoid tumors. Latency III is highly active, strongly stimulating the host immune system. During latency, the EBV selectively expresses a limited set of genes and non-coding RNAs, including six EBNAs (EBNA-1, EBNA-2, EBNA-3A, EBNA-3B, EBNA-3C, and EBNA-LP), three latent membrane proteins (LMP-1, LMP-2A and LMP-2B), and two non-coding RNAs, (EBV-encoded RNAs (EBERs) and microRNAs (miRNAs)) [9]. Among these non-coding RNAs, EBERs (EBER-1 and EBER-2) are consistently and abundantly expressed across all three latency types (I, II, and III) [10], making them highly reliable markers for identifying latent EBV infection in tissue samples through in situ hybridization [11]. In addition, a large cluster of EBV-derived microRNAs originates from *BamHI* A rightward transcripts (BARTs), known as BART miRNAs. These miRNAs are predominantly expressed during latencies I and II [10]. Due to their stability and presence in both tissues and body fluids, BART miRNAs are being investigated as potential noninvasive biomarkers for EBV-related malignancies. However, under certain circumstances, the virus can transition from latency to the lytic phase in B-cells, leading to active viral replication. During the lytic phase, a wide array of EBV genes are expressed to facilitate genome replication and the production of new viral particles [12].

In contrast to the latent state observed in B-cells, the EBV exhibits a preference for undergoing lytic replication within epithelial tissues [13]. Approximately 70 EBV lytic genes are expressed during the synthesis of new viruses, encoding viral structural components that facilitate the alteration of viral morphology and the release of newly produced viruses. The transition from viral latency to productive lytic infection is orchestrated by two genes encoding viral transcription factors, namely *BamHI* Z fragment leftward open reading frame (BZLF1) and *BamHI* R fragment leftward open reading frame (BRLF1) [14]. The expression of these two genes is induced via the cellular differentiation of B-cells or epithelial cells during the latent phase of EBV infection [15]. Upon reactivation into the lytic phase, the expression of early lytic proteins such as *BamHI* M fragment rightward open reading frame (BMRF1), *BamHI* fragment leftward open reading frame 1A (BALF1), and *BamHI* fragment rightward open reading frame (BHRF1) is tightly regulated, promoting replication initiation and triggering EBV genomic DNA replication. This, in turn, leads to the expression of structural proteins, including matrix proteins, capsid components, and envelope proteins. Ultimately, the replicated viral genomic DNA and newly synthesized viral structural proteins assemble into infectious viral particles, released extracellularly through cytosol or cytolysis [16].

### 2.3. Clinical Manifestations

The EBV is implicated in various diseases, including acute mononucleosis, chronic active EBV infection, X-linked lymphoproliferative syndrome (XLP), and B-cell lymphoproliferative disorders. Additionally, EBV infections in epithelial cells can lead to oral hairy leukoplakia, nasopharyngeal carcinoma, and gastric cancer. Reports also link the EBV to smooth muscle sarcoma and other tumor types [17]. A recent study estimated that approximately 298,800 new cancer cases and 173,300 cancer-related deaths worldwide in 2020 were associated with the EBV [18]. Figure 2 illustrates diseases resulting from EBV infection.

#### 2.3.1. Tumor Diseases

Among EBV-associated malignancies, B lymphocyte-derived lymphomas and epithelial-derived carcinomas represent the two major categories, each with distinct patterns of EBV latency and oncogenesis [19]. In B-cell lymphomas, such as Burkitt lymphoma and Hodgkin lymphoma, the EBV typically maintains a latency I or II program [20], with the virus residing episomally in the nucleus and expressing only a limited set of viral genes such as EBNA-1, LMP-1, and LMP-2 to evade immune surveillance. Post-transplant lymphoproliferative disorder (PTLD) is classified as a latency III condition. These malignancies frequently arise under conditions of immune suppression or immune escape, where the inability to clear EBV-infected B-cells promotes clonal proliferation and oncogenic transformation.

In contrast, epithelial-derived cancers, most notably nasopharyngeal carcinoma (NPC) and EBV-associated gastric carcinoma (EBVaGC), exhibit distinct latency programs. EBVaGC is classified as a latency I or I/II condition and does not express LMP-1. However, NPC is classified as latency type II and expresses LMP-1. Unlike B-cell lymphomas where the virus may be present in a subset of tumor cells, EBV infection in epithelial cancers is typically monoclonal and uniformly present, suggesting that it occurs early and contributes to the initiation of transformation. These tumors often express EBNA-1 and latent membrane proteins, particularly LMP-2, thought to modulate signaling pathways, inhibit apoptosis, and alter the tumor microenvironment.

#### 2.3.2. Infectious Diseases

Adolescents are susceptible to developing IM following a primary infection, marked by an incubation period of 4–7 weeks. Symptoms typically include fever, pharyngitis, lymphadenopathy, splenomegaly, and hepatocellular dysfunction, occasionally accompanied by significant jaundice [1]. Patients may experience fatigue, lethargy, and temporary depression. IM frequently leads to prolonged illness in adolescents and middle-aged adults.

In individuals with underlying autoimmune disorders, cancers, inflammatory conditions, and other specific ailments, the reactivation of the EBV can exacerbate the condition, potentially leading to chronic infections and neoplasms [21]. Unlike classic EBV latency in B lymphocytes, CAEBV often involves the infection of T cells or natural killer (NK) cells [22], particularly in East Asian populations. The latency pattern in CAEBV is typically classified as latency II, with the expression of latent genes such as EBNA-1, LMP-1, and LMP-2 [23], depending on the specific cell type infected. Chronic active EBV infection (CAEBV) is a rare condition characterized by severe, persistent, or recurrent IM-like symptoms that manifest following primary EBV infection. Patients typically exhibit a significantly elevated EBV load in peripheral blood, often accompanied by substantial organ damage such as interstitial pneumonia, myelodysplasia, uveitis, hepatitis, and splenomegaly. CAEBV poses significant challenges in terms of treatment, with notably high morbidity and mortality rates.

The deleterious impact of EBV infection on cells induces dysfunction in B-cells, giving rise to XLP [24], a rare, familial, and lethal variant of IM that exhibits a higher incidence in male individuals. Upon EBV infection, patients may develop aplastic anemia and pancytopenia due to hepatic necrosis, followed by the substantial infiltration of cytotoxic T-lymphocytes and cytokine release, potentially resulting in rapid fatality, particularly when concurrent with bacterial or mycobacterial infections.

#### 2.3.3. Immune-Related and Comorbid Disorders

Beyond tumors and overt infections, the EBV may contribute to immune dysregulation and is implicated in several non-malignant chronic conditions, acting as a cofactor rather than a direct cause.

One such condition is chronic fatigue syndrome (CFS), which may follow EBV-related IM [25]. While the exact etiology of CFS remains elusive, EBV reactivation and persistent immune activation have been proposed as contributing factors in a subset of patients.

Furthermore, the EBV has been associated with several autoimmune diseases, including systemic lupus erythematosus (SLE), multiple sclerosis (MS), and rheumatoid arthritis (RA) [26]. Proposed mechanisms include molecular mimicry, epitope spreading, and chronic antigenic stimulation from a latent EBV infection in B-cells.

### 2.4. Diagnostic Significance of EBV Positivity

Although EBV infection is widespread and often asymptomatic, the detection of EBV positivity has nuanced significance in both clinical and research studies. Rather than merely indicating past exposure, EBV positivity often reflects complex interactions that can impact disease progression and experimental outcomes. Understanding the diagnostic implications of EBV status requires carefully considering its role in co-infection, its presence in research samples, and its established links to human malignancies.

In immunocompromised patients and those with underlying chronic conditions, EBV does not act in isolation. It frequently coexists with other pathogens, and its reactivation can contribute to a worsened clinical course. Transplant recipients and individuals with HIV [2], for example, are particularly vulnerable to EBV-associated complications. In such cases, EBV monitoring provides essential information not only about viral dynamics but also about the broader virological environment influencing patient outcomes.

In the laboratory, EBV infection in biological samples can introduce substantial variability into experimental data. Many commonly used cell lines, especially those derived from lymphoid or epithelial tissues, may harbor latent EBV infection without overt cytopathic effects. For example, Sun et al. tested 192 cell lines from the China Center for Type Culture Collection (CCTCC) for the EBV and found that 10 of them were EBV-positive [27]. This latent presence may subtly affect cellular pathways, immune responses, and transcriptional profiles. Unrecognized EBV positivity in research models risks undermining data integrity, particularly in studies exploring virus–host interactions or signaling mechanisms.

The detection of the EBV also holds well-established relevance in the diagnosis and classification of EBV-associated malignancies. In NPC, Hodgkin’s lymphoma, EBVaGC, and extranodal NK/T-cell lymphoma, EBV DNA or RNA serves as both a molecular marker and a potential therapeutic target. Identifying the EBV within tumor tissues aids in confirming diagnosis, guiding treatment decisions, and stratifying patient risk, especially in high-incidence regions where EBV screening may contribute to early intervention strategies.

Furthermore, in managing EBV infection, overcoming a key diagnostic challenge between primary infection and secondary infection is critically important. This distinction is primarily achieved through serological profiling EBV-specific antibodies, often complemented by quantitative nucleic acid detection, as the serological kinetics of EBV-specific antibodies provide important clues for determining the stage of infection. Immunoglobulin G (IgG) antibodies against the viral capsid antigen (VCA), referred to as anti-VCA IgG, persist for an extended, often lifelong period and, therefore, cannot reliably distinguish between recent and past infections. In contrast, immunoglobulin M (IgM) antibodies against VCA (anti-VCA IgM) typically appear during the acute phase and persist for approximately one to two months, serving as a useful marker of recent primary infection. Similarly, IgG antibodies against the early antigen (anti-EA IgG) generally peak around one month after symptom onset and decline within two months, making them another indicator of recent EBV infection [11]. On the other hand, IgG antibodies against the EBNA (anti-EBNA IgG) appear about one month post-infection and persist for life, and they are, therefore, considered a reliable marker of past EBV infection. Thus, primary infection, typically seen in children or adolescents, is characterized by the serological pattern of anti-VCA IgM positivity, anti-VCA IgG positivity, and anti-EBNA IgG negativity, indicating acute-phase infection [28]. In contrast, a profile of anti-VCA IgG and anti-EBNA IgG positivity with anti-VCA IgM negativity usually indicates past infection or viral reactivation, a scenario particularly relevant in immunocompromised patients.

In conclusion, EBV positivity represents far more than an outcome of infection. It is a contextual indicator with diverse applications, ranging from co-infection monitoring to tumor biology and research standardization. Accurately interpreting EBV testing hinges on aligning clinical context, sample characteristics, and disease associations, reinforcing the need for precise and context-dependent diagnostic approaches.

## 3. Diagnostic Methods

The evolution of EBV detection technology can be categorized into three distinct phases (Figure 3):

Before 2000, the main approach involved qualitative methods with limited throughput, relying heavily on histology and clinical inference. This method was primarily utilized for fundamental research and specific pathological diagnoses.

From 2000 to 2019, advancements focused on standardization, automation, and quantification, leading to enhanced detection accuracy and clinical utility. These improvements catered to the growing need for monitoring viral loads in organ transplant recipients and early tumor detection.

Since 2019, a notable transformation in EBV detection technology has occurred. The increasing demand for portable, rapid, and multiplex testing has propelled the rapid progress in isothermal amplification, CRISPR, and other innovative technologies to enable testing across a broader range of samples. Additionally, the integration of digital tools such as AI-assisted analysis and big data amalgamation has started to occur in viral diagnostics. This shift has not only accelerated the speed and precision of EBV testing but has also aligned it more closely with the requirements of primary healthcare and public health surveillance. This marks a new era characterized by convergence and intelligence in the field of EBV diagnostics.

### 3.1. The Last Century—2000: The Basis of Modern Testing Techniques

Since its initial identification in the 1960s [2], varied crucial assays have been successively developed and utilized in the study of the EBV. Over time, there has been a transition from conventional viral morphology and cell culture approaches to adopting molecular biology methods, significantly enhancing both EBV research and clinical diagnostics. Though each technique has its distinct strengths, in practice, they function synergistically, forming a complementary and comprehensive system for EBV research and diagnosis. For example, polymerase chain reaction (PCR) may indicate the presence of viral DNA, while in situ hybridization (ISH) and immunohistochemical staining (IHC) confirm its localization in tumor tissues; serology (via the enzyme-linked immunosorbent assay (ELISA)) may suggest an infection history, while sequencing reveals genotypic characteristics. This integrated toolkit, established by 2000, not only underpins current clinical practices but also continues to support the evolution of rapid and point-of-care EBV diagnostics.

#### 3.1.1. Electron Microscopy (EM)

In the 1970s and 1980s, virology research entered the golden age of microscopic observation. The examination of cytopathic lesions via cell culture and microscopy has been considered a standard method for detecting the proliferation of such viruses. However, it is undeniable that the observation of EBV viruses often requires more sophisticated EM techniques because EBV-infected cells usually do not show significant cytopathic effects (CPEs) for a short period.

In 1964, Anthony Epstein and his team first observed EBV particles via EM after culturing Burkitt’s lymphoma cancer cells derived from an African child. These particles, which were spherical or oval in shape and about 120–200 nm in diameter, could be directly recognized as the EBV using high-resolution images [29]. Subsequently, EM techniques were thought to be useful for identifying replicating viruses infected with intact EBV viroids. Nevertheless, the limitations of this method are obvious, i.e., it is complex and costly and not suitable for routine clinical use.

In the following decades, with the introduction of subsequent molecules and emerging technologies, the use of microscopy alone for EBV detection has been gradually rejected.

#### 3.1.2. Immunofluorescence (IF)

Fluorescence microscopy (FM) has been available since 1941 [30], and its companion IF method quickly became an important tool for EBV detection. In 1968, Volker et al. successfully detected the EBV in cell cultures via the IF technique combined with EM [31]. This method was developed through using fluorescently labeled anti-EBV antibodies (e.g., antibodies against the EBNA and EA, which led to the observation of specific infected cells emitting distinct fluorescent signals under FM, and the method was later used for qualitatively detecting EBV. In 1970, Werner Henle et al. detected, in serum, the presence of antibodies to early EBV antigens via indirect IF [32] to aid the diagnosis of diseases associated with EBV infection, advancing the development of serologic diagnosis.

The IF technique has high specificity and sensitivity due to its ability to visualize the localization and expression of viral antigens in cells, and the method has been used up to now. In recent years, researchers have used it with real-time fluorescence quantitative PCR for the comprehensive diagnosis and condition monitoring of EBV infection, as it can detect serum antibody levels and quantitatively analyze the EBV viral load [33]. It can provide a more comprehensive understanding of the EBV infection status of patients and provide a stronger basis for clinical treatment. In addition, with the development of image recognition and artificial intelligence, the indirect immunofluorescence image analysis method based on machine learning is being optimized, which is expected to improve diagnostic efficiency and automation levels.

#### 3.1.3. In Situ Hybridization (ISH)

In 1969, ISH experiments using radiolabeled DNA probes were first reported by Gall and Pardue [34]. In 1983, ISH began to be applied to detecting DNA fragments of the EBV in the tissues of patients suffering from neoplasms and lymphoproliferative disorders [35], demonstrating its potential for viral localization. This suggests that the use of ISH in EBV detection dates back to at least the early 1980s. In 1991, Coates et al. developed a non-radioactive in situ hybridization technique for detecting EBV in formalin-fixed, paraffin-embedded tissues [36]. This technique utilizes digoxigenin-labeled antisense oligonucleotides that target the small RNA (EBER-1 and EBER-2) sequences encoded by the EBV, greatly improving the sensitivity of the assay.

At the end of the 20th century, EBER ISH was regarded as the “gold standard” method for detecting latent EBV infections in tissue samples [37], and this status remains unchanged today.

#### 3.1.4. ELISA

ELISA has been widely used for immunologically detecting varied viral infections since its introduction in the 1970s [38]. Unlike IF, which requires expensive equipment and cumbersome steps, ELISA is a much simpler method for detecting antibodies to human serum EBV-associated antigens and is widely used in clinical and epidemiological investigations because of its high sensitivity and high throughput.

In 1977, a study compared ELISA with other assays and confirmed that it had high sensitivity in detecting EBV antibodies [39]. In 1985, researchers simplified ELISA and shortened the assay procedure, which further achieved improved assay efficiency [40]. In 1988, a three-step ELISA was proposed [41], which further improved the diagnostic accuracy of nasopharyngeal cancer.

With the maturity of the technology used and the popularity of commercialized kits, ELISA technology has been widely used in the clinical diagnosis of EBV infections. Nowadays, ELISA is used not only to screen for viral antibodies but also to further differentiate between acute, chronic, and latent infections. For example, EBV-specific antibodies detected using ELISA can be related to viral capsid antigens (VCAs), such as VCA-IgM (a marker of acute infection), VCA-IgG (a marker of previous infection), and EBNA-IgG (a marker of latent infection).

#### 3.1.5. PCR

Polymerase chain reaction (PCR) technology was first introduced in 1985 [42] and has rapidly become a central tool in molecular biology and virus detection. The application of this new technique to EBV detection has greatly improved both the sensitivity and specificity of the test. The application of PCR has allowed for detecting EBV to accurately differentiate between latent and acute infections and has enabled the successful detection of the virus in samples with low viral loads.

In 1989, PCR technology was applied for detecting EBV in blood and salivary gland biopsy samples [43], indicating that the application of PCR technology in EBV detection can be traced back to at least the late 1980s. Since then, PCR technology has been widely used in the clinical diagnosis of EBV-related diseases, especially in monitoring chronic lymphocytic leukemia and lymphoma.

By the 1990s, PCR technology was evolving, and EBV-related research was proliferating. On one hand, researchers began to experiment with combining PCR with other detection methods. For example, PCR-synthesized probes were used for the ISH of EBER in 1995 [44], and the combined PCR-ELISA was developed in 1997 [45]. On the other hand, with innovations in PCR technology, researchers have successively used the reverse transcription-polymerase chain reaction (RT-PCR) [46], quantitative competitive PCR (QC-PCR) [47], nested multiplex PCR [48], modified in situ polymerase chain reaction (IS-PCR) [49], semi-quantitative PCR [50], fluorescent quantitative PCR [51], multiplex PCR [52,53], and reverse transcription multiplex PCR [54] techniques to detect the EBV.

PCR techniques have been an important detection tool to date. These techniques not only improve the sensitivity of EBV detection but also provide strong technical support for viral load monitoring, typing analysis, and clinical diagnosis.

#### 3.1.6. Sequencing Technology

In 1977, Walter Gilbert and Frederick Sanger invented the first sequencer [55], of which the sequencing method invented by Sanger known as the first-generation sequencing technology. In 1984, scientists completed the first whole-genome sequencing of the EBV B95-8 strain [56], representing an important sequencing technology for EBV research applications.

Although the Sanger method has the advantages of easy operation and high accuracy, the sequence length of each read is limited (about 700–1000 bases), making it difficult to meet the demand of large-scale genome research. With the development of second- and third-generation HTS technologies, modern EBV research and clinical typing are increasingly relying on new sequencing platforms, but the Sanger method still has irreplaceable value in sequencing and validating specific target regions.

#### 3.1.7. Immunohistochemical Staining (IHC)

IHC is a technique that combines antibodies with tissue sections and is widely used in diagnosing viral infections. IHC can detect viral antigens directly in tissue samples, which makes it particularly suitable for histologic studies for the localization and distribution analysis of viral infections.

The application of this method for EBV detection dates back to at least 1978, when Kurstak et al. demonstrated the high sensitivity of the method using an indirect immunoperoxidase technique for detecting EBV antigens, early antigens (EAs), viral capsid antigens (VCAs), and nuclear antigens [57]. In 1989, some researchers used an affinity-biotin complex (ABC) immunocytochemical technique to detect IgA antibody levels for EBV viral capsid antigens and early antigens (EAs) in the sera of nasopharyngeal carcinoma (NPC) patients and demonstrated that the ABC technique is a more sensitive and comparably specific assay [58]. As the study of EBV viral antigens intensified, scientists discovered more EBV-related specific antigens. For example, in 1995, it was shown that the expression of LMP-1 in tumor cells was closely related to EBV infection [59]. By using anti-EBV antibodies in combination with tissue sections, this method can directly localize and identify EBV antigens at the histological level, greatly facilitating the study of the EBV in tumors such as Hodgkin’s lymphoma [59], IM [60], and other EBV-related tumors.

In the following decades, IHC has been widely used in the typing and early diagnosis of EBV-related tumors. However, in the Chinese Society of Clinical Oncology Guidelines for the Diagnosis and Treatment of Lymphoma, it is stated that the sensitivity of IHC is often inferior to that of in situ hybridization, but IHC still plays a unique role in displaying the expression of viral proteins and the relationship with histiocytes, and it is one of the most commonly used assays in the pathological diagnosis of lymphoma [61].

### 3.2. 2000–2019: The Proposition of Innovative Detection Methods

Between 2000 and 2019, before the emergence of COVID-19, the field of EBV detection underwent a transformative phase marked by the rapid advancement of molecular biology techniques. Notably, methods such as microarray technology, high-throughput screening, and isothermal amplification emerged as prominent modalities for EBV detection. This era was defined by the diversification, enhanced efficiency, and precise clinical utilization of detection methodologies.

#### 3.2.1. Microarray Technology

The concept of microarray technology was first proposed in 1983. Subsequently, the laboratories of Stanford University first published a study on gene expression profiling microarrays in *Science* in 1995 [62], and DNA microarrays (also known as Biochips, or Genechip) have entered a new stage of development and are widely used in gene expression analysis and genomics research. Microarray technology has reached a new stage of development and is now widely used in gene expression analysis and genomics research, with protein and RNA microarrays representing important branches of this technology.

By 2002, researchers had already investigated the expression of interleukin-1β in gastric cancer caused by EBV infection [63], marking a traceable application of microarray technology in EBV detection. Through high-throughput detection, researchers can rapidly screen different EBV genotypes and study the relationship between viral variants and pathogenicity. Some researchers have used microarray technology to study the changes in gene expression in B lymphocytes following LMP-2A protein expression in the EBV [64]. In addition, microarray technology has been used for viral surveillance in large-scale populations, especially for detecting multiple herpesviruses, to provide a scientific basis for public health policies. For example, in 2004, gene chip technology was used to detect multiple human herpesviruses [65]. In the same year, a novel protein microarray technology that can be used to simultaneously detect the presence of antibodies against multiple pathogens (e.g., hepatitis C virus, hepatitis B virus, human immunodeficiency virus, and EBV) was reported [66].

With advancements in technology, the combined use of microarray technology with other detection techniques, such as multiplex PCR [67,68], VINAray technology [69], and multiplex RT-PCR [70], emerged for the comprehensive detection of multiple viruses, including the EBV. This approach has become a crucial high-throughput screening tool in virology research and clinical diagnosis.

Today, microarray technology continues to offer advantages in specific applications targeting EBV detection, especially in processing large sample numbers, genotyping, and DNA methylation studies. However, with the development of technology, it is gradually being replaced by these techniques in gene expression analysis, transcriptome analysis, and whole-genome sequencing.

#### 3.2.2. High-Throughput Sequencing (HTS)

Before the invention of HTS, late-20th-century researchers typically relied on first-generation sequencing technologies to sequence the EBV. For example, a 2005 study employed shotgun sequencing to analyze the genome sequence of the EBV GD1 strain in nasopharyngeal carcinoma patients [71]. However, these early sequencing methods had limitations, such as the need to enrich DNA samples by culturing EBV-transformed cell lines or PCR amplification.

Fortunately, the development of HTS in 2005 revolutionized traditional Sanger sequencing. It effectively addressed the limitation of single sequence measurements per run by enabling the simultaneous analysis of hundreds of thousands to millions of nucleic acid molecules in a single operation. As such, it is also known as next-generation sequencing (NGS) or second-generation sequencing.

The emergence of HTS technology has brought the genome study of the EBV into a new stage. HTS technology can sequence the entire viral genome quickly and with high precision, and it can identify small mutations and variations, providing new perspectives on the diversity of viruses and their pathogenic mechanisms. In 2011, Liu et al. performed deep sequencing of the whole EBV genome extracted from tumor tissues of NPC patients using a study with HTS technology [72]. This study successfully assembled a clinically isolated EBV genome with a size of 164.7 kb, laying the foundations for clinical EBV load monitoring and variant analysis. Since then, researchers have further utilized HTS technology to sequence EBV genomes in different cells or tissues, including mycoplasma infection-induced peripheral blood B lymphocyte cell lines [73] and lung cancer biopsy samples [74]. Meanwhile, this sequencing method has also been applied to detecting varied virally complex biological samples, including the EBV, demonstrating the promising applications of HTS technology in multiviral detection [75].

In recent years, single-cell sequencing, an advanced application of HTS, has enabled researchers to analyze EBV infection at the individual cell level. This approach allows for identifying viral gene expression patterns, latency types, and host–virus interactions within heterogeneous cell populations [76]. For example, studies have applied single-cell RNA sequencing to characterize EBV gene expression dynamics in tumor-infiltrating B-cells and explore how the EBV shapes the tumor microenvironment at the single-cell level [77]. These developments offer new insights into the complexity of EBV-associated pathogenesis and potential immune evasion strategies.

At present, the birth of third-generation sequencing technology, also known as single-molecule sequencing technology, enables the HTS of long nucleic acid fragments from end to end, providing direct reads of sequences tens of thousands of bases in length. However, this technology currently has certain limitations in terms of throughput, cost, and data accuracy, and it has not yet reached the breadth of application of second-generation sequencing technology. Therefore, HTS is still the main sequencing method used for EBV detection.

#### 3.2.3. Isothermal Amplification

Isothermal amplification technology is a class of techniques that amplify nucleic acids at a constant temperature with the advantages of speed, simplicity, and the absence of complex equipment. Some isothermal amplification techniques have been invented since the last few years of the last century, with Nucleic Acid Sequence-Based Amplification (NASBA) proposed in the early 1990s [78], Strand Displacement Amplification (SDA) first proposed by Walker et al. in 1992 [79], Rolling Circle Amplification (RCA) proposed in 1998 [80], Loop-mediated Isothermal Amplification (LAMP) proposed by Japanese scholar Notomi in 2000 [81], and Recombinase Polymerase Amplification (RPA) first proposed in 2006 [82]. However, the wide application of isothermal technology to molecular diagnosis, disease detection, and other fields mainly occurred after 2000.

Isothermal amplification assays for EBV date back to 1998, when studies were conducted to develop a NASBA assay based on the amplification of nucleic acid sequences [83]. In 2001, researchers applied ramification amplification (RAM), a technique used for Raji cells for detecting EBV sequences [84]. In the following period, the LAMP technique became the most commonly used isothermal amplification technique and was shown to be comparable to real-time PCR in terms of sensitivity and specificity in serum detection in EBV patients. To improve the accuracy of the LAMP technique in clinical applications, some improvements have been made by researchers. For example, RT-LAMP was utilized to assess the expression of latent versus lytic genes during EBV infection [85]. The low cost and ease of use of the LAMP technique, due to its ability to rapidly and highly sensitively detect EBV DNA without complex equipment, especially in resource-limited areas, make it a promising tool applicable to EBV detection.

Today, isothermal amplification remains a convenient, rapid, and effective means of targeting the EBV. Several companies and research institutes have developed isothermal amplification-based EBV assay kits, which have found initial application in clinical diagnostics. For example, Sun et al. performed quality control of EBV contamination in cell banks by employing RPA technology [27].

### 3.3. 2019–Now: Multidisciplinary Application in the Post-COVID-19 Era

The COVID-19 pandemic led to a significant increase in the worldwide need for swift, precise, and effective viral detection tools. This surge notably accelerated advancements in molecular biology and immunology testing technologies. Throughout the pandemic, viral detection encountered substantial obstacles related to speed, sensitivity, and scalability for large-scale screening. As a result, the detection methodologies for the EBV were both influenced and enhanced, thereby fostering further advancements in viral detection techniques.

#### 3.3.1. CRISPR-Cas Technology

In recent years, the gene editing technology CRISPR has helped researchers to make significant progress in the field of virus detection. The CRISPR/Cas system is an immune defense system consisting of a combination of Cas effector proteins and CRISPR RNAs (crRNAs). Researchers have discovered that some Cas proteins (e.g., Cas12, Cas13, etc.) have “collateral cleavage” activity, which means that Cas nuclease can cleave fluorescent reporter molecules, and they have been applied to biosensing and developed into effective, rapid, low-cost and highly sensitive molecular detection methods.

In particular, the epidemic has further fueled the development of this technology. For example, researchers explored the potential of the CRISPR-Cas system in screening for SARS-CoV-2 therapeutic targets [86]. In 2022, a team investigated and developed a CRISPR-LbCas12a-based rapid detection tool [87], which combines rapidity, sensitivity, and accuracy and can detect SARS-CoV-2 in 20 min with a sensitivity comparable to RT-PCR. However, Cas9 cannot visualize the detection results due to the lack of trans- cleavage activity, and it is usually necessary to combine it with other methods for virus detection. For example, Moon’s team [88] and Xiong’s team [89] established a platform to detect viruses by using a color reaction and colloidal gold test strips in conjunction with the CRISPR/Cas9 system, respectively.

CRISPR-Cas systems are categorized into two main classes and six types based on the composition and mechanism of action of effector proteins [90]. Among them, Class 1 systems are composed of multiple proteins used to perform functions in complexes, with complex structures, and they are mainly used in basic research but have not yet been widely used in virus detection. The Class 2 system, which is centered on a single Cas protein, has the advantages of a simple structure and convenient operation and has become the mainstream choice in the field of rapid virus detection. In the Class 2 system, different types of Cas proteins are suitable for different detection needs due to the differences in their targeting molecules and cleavage mechanisms, among which the most commonly used nucleic acid detection kits are Cas9, Cas12, and Cas13. The major difference between these three proteins is that the Cas9 protein only cleaves dsDNA, while the Cas12 protein can cleave ssDNA non-specifically, Cas13 protein can cut specific RNA. Their respective advantages and disadvantages are shown in Table 1.

Since the first successful use of CRISPR-Cas9 for gene editing in mouse and human cells [26], this powerful gene editing tool has been applied in varied fields. Cas9 can accurately recognize and cleave specific double-stranded DNA sequences, but it does not possess trans-cleavage activity itself, so this system is less commonly used in EBV detection. Current studies have mainly applied it to gene editing and cloning the EBV, for example, one study reported that Yuen et al. used CRISPR/Cas9 technology to achieve targeted editing in several human epithelial cell lines latently infected with the EBV [91]. In 2016, it was shown that CRISPR/Cas9-mediated free DNA cleavage of the EBV significantly improved the cloning efficiency of disease-associated viral strains [92].

In 2018, the CRISPR-Cas12a-based DETECTR was introduced [93]. Cas12 activates non-specific single-stranded DNA cleavage activity upon recognizing a specific dsDNA target, releasing a pre-designed fluorescent or colorimetric signaling probe, and it is commonly used in conjunction with isothermal amplification techniques (e.g., RPA) to achieve highly sensitive detection of EBV DNA. Li et al. combined RPA with CRISPR-Cas 12a to realize rapid and portable detection of the EBV [94]. The CRISPR-Cas system has also been used in diagnosing and dynamically monitoring EBV-associated tumors such as nasopharyngeal carcinoma [95].

A CRISPR-based Cas13-based viral detection technology, SHERLOCK, was also introduced in 2018 [96]. Cas13, on the other hand, specifically recognizes single-stranded RNA and triggers non-specific ssRNA cleavage, which is suitable for detecting EBV expression products and is particularly suitable for determining the active transcriptional state of the virus or the expression profile of latent infection. In 2019, studies were conducted to detect EBV DNA in plasma using CRISPR-Cas13 at room temperature [97]. To reduce the risk of contamination caused by multi-step uncapping, a study proposed an isothermal one-pot RNA detection platform called the Simultaneous Amplification and Testing platform based on Cas13a, which was successfully applied to 23 EBV-infected samples [98]. In addition, a study developed the CRISPR-Cas13a Gemini system, which enables the highly sensitive detection of discontinuous target RNAs such as EBER-1 and EBER-2 [99].

In summary, the different types of Cas proteins in the Class 2 system have their own advantages in EBV detection applications based on the differences in their targeting molecules and signal release mechanisms, and they can support multilevel diagnostic needs ranging from genome identification to expression level monitoring. The programmability and signal amplification capability of this system make it an important tool for developing a highly sensitive, low-cost, on-site EBV detection platform.

#### 3.3.2. Artificial Intelligence (AI)

The concept of AI was first introduced in 1956 [100]. It emphasizes the design of intelligent systems that can learn from data and make decisions and predictions accordingly. In recent years, AI has played a significant role in several of these due to the large amount of data generated from sources such as evolving high-resolution medical imaging technology, genome sequencing, and electronic medical records. On one hand, AI can identify characteristic patterns of viruses by analyzing a large number of viral genome sequences, clinical sample data, or cell images. For example, a research team successfully identified 161,979 potential RNA virus species and 180 RNA virus supergroups in 10,487 macro-transcriptomes worldwide by integrating AI algorithms (LucaProt) with sequence and predicted structural information [101]. On the other hand, AI can also monitor viral activities in host cells in real time and detect early signs of viral infection on time by analyzing cell images or gene expression data. For example, the AINU tool developed by Southern Medical University can detect small changes in the nuclei of cells within one hour of viral infection, thus enabling early diagnosis [102].

After the COVID-19 outbreak, the application of AI and big data technologies in virus detection has been gradually emphasized. AI technologies have been used in several aspects of SARS-CoV-2 detection, including viral sequence analysis [103], molecular virus detection [104], and viral image recognition [105]. These techniques have gradually developed and matured in COVID-19 detection and positively impacted the detection of the EBV, especially in the two important areas of machine learning (ML) and deep learning (DL).

ML, as one of the core branches of AI, emphasizes that models learn patterns and perform prediction tasks through training data. In EBV detection, ML is commonly used for biomarker-based classification modeling. The analysis of result data and related quality control in existing EBV assays often still needs to be carried out manually. However, the diagnostic process may suffer from diagnostic errors, wasted resources, and inefficiency, so there is an urgent need for algorithms to help. In 2024, a research team developed PCR.ai, an AI-based analysis technology specifically designed to automate and standardize real-time fluorescent quantitative PCR (qPCR) assay results. The technology was applied to varied virus detection scenarios, including EBV detection [106]. Validated EBVs were detected and quantified using both manual routine analytical methods and PCR.Ai with 100% compliance. In addition, a study developed six ML models for EBV-associated gastric cancer diagnosis, including logistic regression, extreme gradient boosting, random forest, support vector machine, Gaussian naive Bayes, and the K-nearest neighbor algorithm. The results showed that the RF model performed the best [107].

DL, as a subfield of ML, is based on a multi-layer neural network structure, which can automatically extract complex features from raw data, and is especially suitable for high-dimensional unstructured data such as medical images and genome sequences. DL is more often applied to tasks such as pathology image recognition and sample pattern mining in EBV detection. Some of the major DL methods include Convolutional Neural Networks (CNNs), Recurrent Neural Networks (RNNs), and Long Short-Term Memory (LSTMs) [108]. Contemporary studies of the EBV mainly use CNN methods. For example, CNNs have been applied to establish automated detection platforms for EBV-positive and microsatellite instability, mismatch repair defective tumors [109], and gastric cancer [110]. The application of AI technology in EBV-related tumor detection is particularly significant. In 2019, it was shown that deep learning-based PET/CT radiomics can effectively predict the prognosis of patients with advanced nasopharyngeal carcinoma [111]. In 2021, a deep-learning classifier for detecting EBV in gastric cancer tissue sections was developed and validated [112]. In the same year, another EBV infection prediction algorithm for H&E-stained histopathologic sections of gastric cancer was developed. Notably, this prediction categorized patients into EBV infection categories that were significantly associated with prognosis [113]. In addition, DL has been used in EBV integration site prediction studies. Researchers developed DeepEBV, a deep-learning model that automatically identifies local genomic features and predicts viral integration sites [114].

Overall, the process of AI application in the field of EBV detection can be roughly divided into the following steps: sample reception, analysis, and outputs (Figure 4). AI technology is playing an increasingly important role in improving the sensitivity, specificity, and automation of viral testing and opening up new directions for detecting many viruses, including the EBV.

#### 3.3.3. Point-of-Care Testing (POCT)

In recent years, POCT equipment has developed rapidly, driven by both technological innovation and public health needs. POCT is an emerging technology for performing rapid medical testing at the patient’s side, which realizes on-site sampling and immediate analysis with portable equipment and reagents [115]. Compared with traditional testing methods, POCT is faster, more efficient, and less costly. In the field of virus detection, this method is designed and rationalized to achieve fast, convenient, and accurate on-site detection. These devices often combine varied advanced detection technologies and miniaturized designs, such as optical detection, electrochemical detection, biosensor technology, and isothermal amplification, to meet the needs of different application situations.

In 2003, the World Health Organization (WHO) introduced a set of standards for desirable medical testing techniques, the ASSURED standards, which have become the benchmark for the wide acceptance of POCT techniques [116]. With technological advances and future needs evolving, in 2019, some experts further supplemented the two standards of real-time connectivity and ease of specimen collection, which evolved into the WHO REASSURED medical testing standard [117,118].

The 2019 COVID-19 outbreak fueled research in this area. In response to demand for large-scale screening, rapid portable testing platforms such as the CRISPR-Cas system combined with isothermal amplification technology have emerged, and a large number of commercialized products have been rapidly deployed and widely used around the world. For example, one study proposed an applied portable instrument that can rapidly and reliably detect SARS-CoV-2 infection in any environment [119].

The technological spillover from the COVID-19 outbreak has also significantly contributed to the expansion of portable devices in other areas of viral detection, especially showing potential in the rapid detection of tumor markers associated with chronic infections such as the EBV. One study even developed a device that can detect DNA from both SARS-CoV-2 and the EBV to help validate patients with mixed EBV/SARS-CoV-2 infections [120]. According to its different detection principles and technology platforms, POCT applied to EBV detection mainly focuses on immunochromatographic and molecular diagnostic categories.

Immunochromatographic POCT is based on the antigen–antibody-specific binding reaction, and common forms include colloidal gold test strips and fluorescent immunochromatography, mainly used for detecting pathogen antigens or host antibodies. The technology is low-cost, is easy to operate, has intuitive results, and is widely used for the initial screening and rapid on-site judgment of infectious diseases. For example, one study established an assay utilizing the detection of anti-EBNA-IgG and anti-EBNA-IgA, which ensures the accuracy of on-site testing for the screening, diagnosis, and surveillance of NPC patients [121].

Molecular diagnostic POCT uses pathogen nucleic acid as the detection target and uses constant temperature amplification technology (e.g., LAMP, RPA) for detection, which has high sensitivity and specificity and is suitable for early infection detection. After 2021, several research teams began to experiment with porting SARS-CoV-2 portable platforms for the targeted detection of the EBV, using LAMP technology to detect genes such as EBNA-1, LMP-1, *BamHI*-W, and other genes or coupling the CRISPR-Cas12a system with isothermal amplification reactions to enable the on-site detection of EBV nucleic acids via test strips. In 2022, a team of researchers developed a handheld nano-centrifugal device for EBV detection that is cost-effective (as low as USD 0.13) and capable of spinning six samples simultaneously. By combining the CRISPR/Cas12a system with an isothermal amplification reaction, it is possible to control the entire operation of a virus assay in less than one hour [122]. Some studies have achieved detection sensitivities as low as a single copy/µL, such as the study that proposed a molecular assay combining MCDA and LFB for the rapid, simple, and sensitive specific detection of EBV, known as EBV-MCDA-LFB, which integrates multiple cross displacement amplification (MCDA) with nanoparticle-based lateral flow (LFB), with a sensitivity of up to seven copies per reaction, which can be accomplished in less than 60 min [123]. In 2024, a study was conducted to apply ring-primed fluorescence for the first time to the self-bursting ring-mediated isothermal amplification (FLOS-LAMP) technique, which established a novel multiplex assay for detecting EBV and B19V [124].

Besides applying the above two main types of testing methods, some teams have applied other techniques such as highly sensitive label-free electrochemical immunosensor for detecting the EBV [125].

Therefore, the development path of portable testing equipment clearly shows a COVID-19 test for promoting the standardization of platforms and then expanding to other pathogens [126]. Regarding the trend of technology migration, EBV testing is one of the important areas to benefit. Despite rapid technological progress, most EBV-related POCT platforms are still in the research or prototype stage. To date, no EBV POCT assay has been officially approved for routine clinical use. Their use remains largely unstandardized, and these tests are not widely adopted in tertiary hospitals. Molecular POCT devices for EBV DNA detection, such as LAMP- or CRISPR-based platforms, remain experimental and have not yet received clinical approval in any country.

#### 3.3.4. Multi-Omics Analysis

The COVID-19 pandemic has driven the widespread application of high-throughput multi-omics technologies in virology. Many research teams have carried out multi-omics data collection, including genomics, transcriptomics, proteomics, metabolomics, and immunophenotyping, to deeply analyze the relationship between viral reactivation and host immune response. Clinical studies have also further validated the link between COVID-19 and EBV reactivation. For example, a comprehensive multi-omics analysis of reactivation of known chronically infected viruses, including the EBV, in patients with neococcal pneumonia was performed, and the data provided new immune, transcriptional, and metabolic biomarkers of viral reactivation [127]. The multi-omics approach has made possible simultaneously detecting viral and host immune markers and accelerated the systematic development of EBV research.

As early as 2019, a team of researchers used transcriptomics, metabolomics, and lipidomics to comprehensively investigate the abnormal metabolism of EBV-associated gastric cancers [128]. In 2024, researchers used GC-MS data to screen for a new biomarker associated with EBV infection, improving response prediction to gastric cancer immunotherapy [129]. In the same year, a genome-wide multi-omics approach was also used to investigate the link between the EBV and periodontitis [130]. In 2025, Yang et al. investigated the molecular characterization of EBV-associated primary lymphoepithelial carcinoma of the lungs using whole-exome sequencing (WES) and whole-transcriptome sequencing of RNA (RNA-seq) techniques [131].

For nasopharyngeal carcinoma (NPC), an important disease caused by the EBV, the application of multi-omics is even more abundant. In 2025, a research team used the DNA methylome and transcriptome in combination with single-nucleus RNA sequencing to study DNA methylation and gene expression patterns in NPC and tumor stem cells [132]. This study identified aberrant DNA methylation-related biomarkers, including nine NPC-specific diagnostic markers. In the same year, Zeng et al. conducted the largest NPC whole-exome sequencing association study to date, including 6969 NPC cases and 7100 controls [133]. Using bulk transcriptomics and single-cell RNA sequencing data in combination with experimental validation, they demonstrated that the RPL14 variant regulates the EBV life cycle and NPC pathogenesis. It opens up new avenues for personalized risk assessment, early diagnosis, and targeted therapy for NPC.

These findings provide direction for applying multi-omics approaches in EBV detection, prompting researchers to begin systematically correlating EBV transcripts, viral proteins, and other biomarkers with host immune status. In the post-COVID-19 pandemic era, multi-omics technologies are driving EBV research into a high-throughput, integrated phase of virus–host interactions.

### 3.4. Summary

In conclusion, varied EBV detection methods have been developed from the last century to the present. The efficacy of these assays differs based on the sample type, processing techniques used, sensitivity, and cost, as detailed in Table 2.

Conventional methods for detecting EBV are commonly utilized. However, some techniques have specific limitations, such as PCR, which, though highly sensitive, often requires costly equipment and trained personnel; ELISA is cost-effective and scalable but may suffer from cross-reactivity. The COVID-19 pandemic catalyzed the adoption of portable testing devices, isothermal amplification techniques, and CRISPR-Cas technology for expedited on-site testing. Moreover, integrating AI algorithms with HTS and multi-omics analysis presents a more precise and comprehensive approach to EBV detection. To enhance the guidance provided in this review, this article clearly distinguishes between the clinical or scientific research uses corresponding to different EBV assays (Figure 5). Future advancements in EBV detection are anticipated to prioritize affordability, heightened sensitivity, rapidity, and convenience, particularly for applications in extensive screening initiatives and resource-constrained settings, thereby holding promise for widespread utilization.

## 4. Future Prospects

Reviewing the development of EBV detection technology, its technological evolution has been significantly influenced by the contemporaneous context, the needs of infectious disease prevention and control, and the influence of interdisciplinary technological integration. From the traditional methods based on immunology and cell culture in the last century through the rise of molecular detection methods such as PCR and real-time fluorescence quantification in the 2000s to the rapid development of the CRISPR-Cas system, AI algorithms, and portable detection platforms in the last decade, EBV detection systems have experienced a multidimensional evolution from sensitivity and throughput to portability and intelligence.

Driven by the technological spillover effect of COVID-19, virus detection approaches are facing the new requirements of being faster, more accurate, and more scenario-based [126], and EBV testing has begun to shift toward point-of-care and single-copy sensitivity. In rapid screening, isothermal amplification strategies represented by the RPA, LAMP, and MCDA, coupled with the CRISPR system, significantly decrease the reaction time and reduce dependence on instruments, making them suitable for large-scale pre-screening or grassroot deployment. In terms of precision diagnosis, the introduction of ddPCR, AI-assisted image recognition, and deep learning integrated locus analysis technologies has significantly improved the EBV detection rate of in the early stage of tumor development and the latent stage, potentially enabling individualized intervention.

Although the current technology has realized the initial transition from the laboratory to the clinic, it still faces three bottlenecks: First, the EBV load in the latent infection stage is extremely low, and it is necessary to develop a combination of markers with more amplification efficiency and specificity. Second, the spectrum of EBV-associated diseases is broad, including nasopharyngeal cancer, gastric cancer, lymphoma, etc., and the detection targets, sample types, and sensitivity requirements of different diseases differ significantly, while the detection platform lacks uniformity. Third, there is still a gap between the standardization of field rapid screening devices and laboratory platforms in terms of contamination control and quantitative capacity.

In addition, while COVID-19 has accelerated the evolution of intelligence regarding EBV testing, there is still much room for advances in the approaches for digitizing and managing the testing data associated with this virus. In recent years, blockchain has emerged as a key digital infrastructure alongside the Internet of Things and AI. During epidemics, data trustworthiness and improving the efficiency of research collaboration are very important, and the decentralized, untamperable, and traceable features of blockchain technology make it very suitable for managing viral detection data. In 2021, a study proposed a low-cost blockchain and AI-coupled self-testing and tracking system for neococcal pneumonia and other emerging infectious diseases [134]. At this stage, the application of this technology to the EBV is rare. The future integration of EBV testing data (viral load, antibody titer, multi-omics characterization) into the blockchain system has some potential advantages. In particular, noninvasive testing technologies are promising for use in virus detection at the clinical stage. Noninvasive testing technologies made significant progress during the COVID-19 pandemic, especially for rapid screening and reducing the risk of transmission. For example, Yao’s team at Peking University developed a noninvasive screening system for SARS-CoV-2 infection based on exhaled breath volatile organic compounds (VOCs) [135]. The system can identify 12 key VOC markers in the exhaled breath of patients with a specificity and a sensitivity of more than 95%. A research team from Nanyang Technological University in Singapore developed a surface-enhanced Raman scattering-based breath analysis module that can screen for SARS-CoV-2 in less than 5 min [136]. The successful application of these techniques provides valuable insights and implications for EBV detection. Especially in diagnosing and treatment monitoring NPC, this technology can achieve real-time monitoring of patients’ treatment responsiveness and risk of recurrence and metastasis, as well as provide support for individualized treatment. A research team successfully achieved the real-time monitoring of patients’ treatment responsiveness and risk of recurrence and metastasis by dynamically tracking circulating tumor-derived EBV DNA (ctEBV DNA) in the plasma of nasopharyngeal cancer patients [137]. Despite the advantages of noninvasive testing techniques in EBV detection, there are still some challenges, such as the standardization of testing methods and mutual recognition of results. Further research and optimization of the testing process are needed in the future to improve the universality and reliability of the test results.

In conclusion, EBV testing is gradually moving from “detection of infection” to “hierarchical diagnosis and risk assessment”, and future development will not only represent a technological breakthrough but also a comprehensive change in multidisciplinary synergy, the intelligent integration of data, and clinical scenario orientation. Driven by rapid screening and accurate diagnosis, EBV testing is expected to play a more critical role in the prevention and control of infectious diseases and the individualized treatment of tumors. This will involve exploring the role of personalized medicine, potential joint screenings with other diseases, and interdisciplinary collaborations that extend beyond scientific fields to influence public health policy-making, thus driving EBV research forward through concerted efforts by diverse professional disciplines.

## Figures and Tables

**Figure 1 viruses-17-01026-f001:**
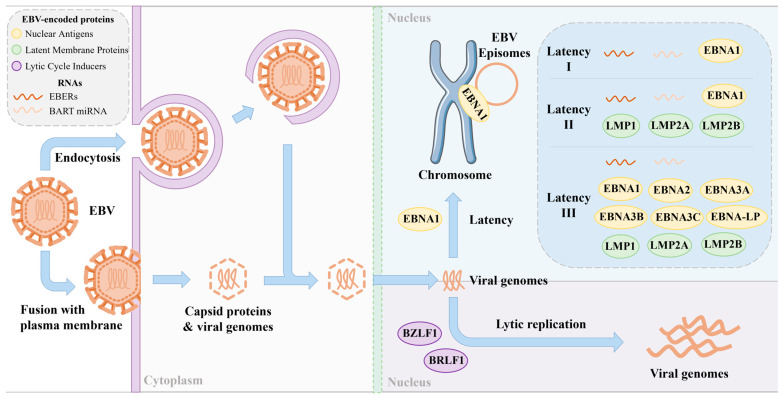
EBV infection and protein expression during latent and replicative phases. EBV latency can be categorized into three types: latency I, latency II, and latency III. Each latency phase produces a limited and unique set of viral proteins and RNAs.

**Figure 2 viruses-17-01026-f002:**
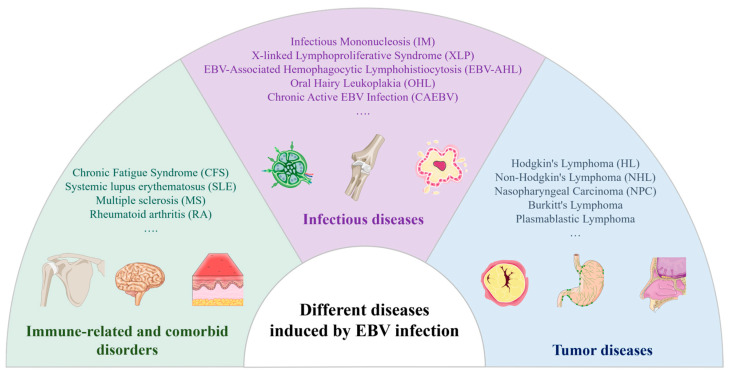
Diseases induced via EBV infection.

**Figure 3 viruses-17-01026-f003:**
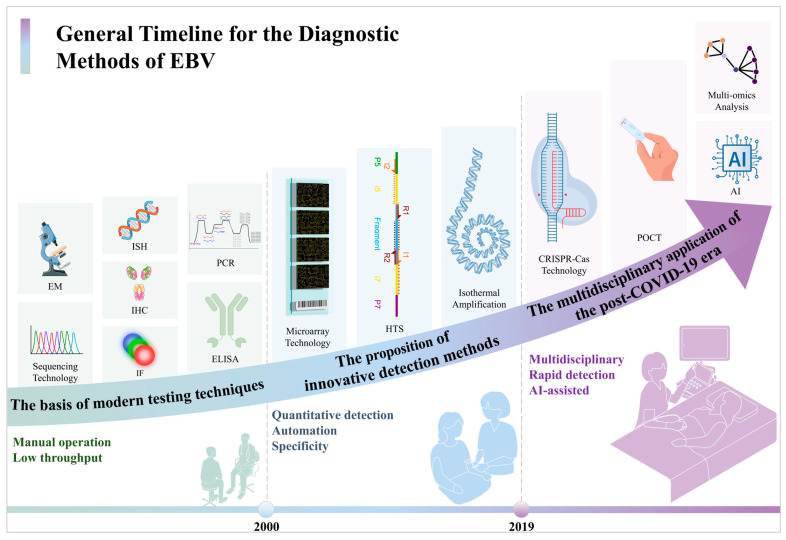
The general timeline for the diagnostic methods for the EBV.

**Figure 4 viruses-17-01026-f004:**
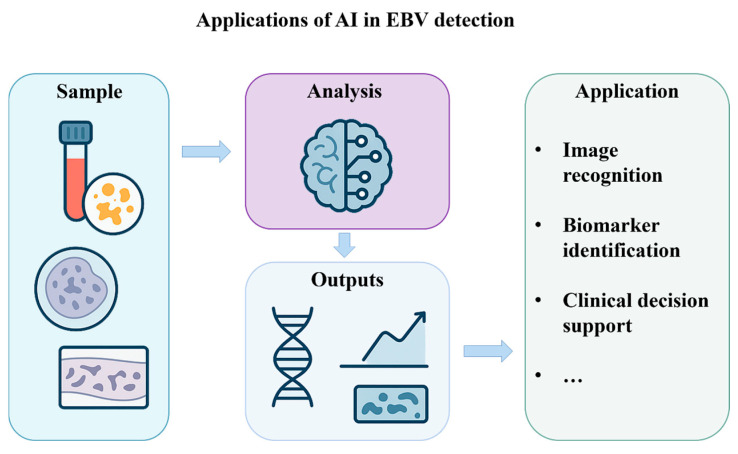
Applications of artificial intelligence in EBV detection.

**Figure 5 viruses-17-01026-f005:**
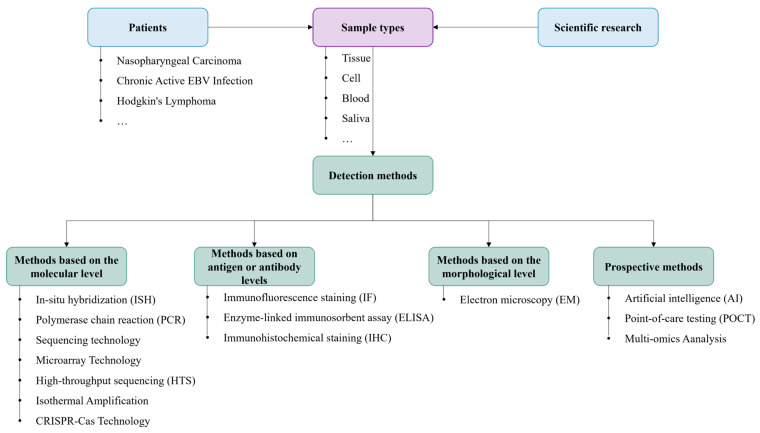
A flow chart of different categories of EBV tests.

**Table 1 viruses-17-01026-t001:** Advantages and disadvantages of Cas9, Cas12, and Cas13.

Types	Advantages	Disadvantages
Cas9	Discovered earlier and well-developed; sgRNA is synthesized from crRNA and tracrRNA, which is more accurate for locating the PAM sites of target genes.	Only dsDNA could be cleaved, so the detection range is limited; No trans-cleavage activity, so it is difficult to detect in vitro.
Cas12	Only crRNA binding to Cas12 protein is required for the method, which is not difficult to detect; Trans-cleavage activity, this type could be independently detected in vitro.	PAM site sequence is TTTV (V is A, G or C), so crRNA design is difficult; Only dsDNA and sSDNA can be cleaved, and RNA cannot be recognized and detected.
Cas13	Having a PFS sequence comparable in action to the PAM sequence, which is composed of sequences located at the 3′ end of the spacer sequence consisting of an A, U, or C, thus increasing the error tolerance of the Cas13 protein; Tolerates single base mismatches between rRNA and target sequences without affecting cleavage efficiency; Trans-cleavage activity, this type could be independently detected in vitro.	Only RNA can be cleaved, and if DNA detection is required, DNA needs to be transcribed into RNA in vitro and then detected, which increases the risk of contamination in the experimental process.

**Table 2 viruses-17-01026-t002:** A comparison of different testing methods for the EBV.

Methods	Samples	Sample Preparation	Applicable Environment	Sensitivity	Cost	Advantages	Disadvantages
EM	Tissue, cells	Samples need to be fixed and sectioned into thin slices; requires high-end electron microscopy equipment	Laboratory	Low, but can observe viral morphology	High, expensive equipment and complex operation	Can directly observe virus morphology, suitable for virology studies	Complex operation, expensive equipment, low sensitivity, limited application range
IF	Tissue, cells	Samples need to be fixed, stained, and labeled with specific antibodies	Laboratory	Moderate, usually lower than PCR	High, expensive equipment and complex operation	Results are visualized, suitable for qualitative and quantitative detection of specific antigens	Lower sensitivity, background interference may affect results, antibody specificity required, quantitative analysis may require advanced image processing software and rigorous standardization
ISH	Tissue, cells	Samples need to be fixed, cleaned, and hybridized with probes targeting specific RNA or DNA	Laboratory	High, detects specific viral RNA/DNA	High, depends on lab equipment and reagents	Allows localization of viral gene expression in tissue sections, high sensitivity	Complex operation, long sample preparation time, limited application
ELISA	Serum, plasma, tissue	Samples need to be extracted and added to a plate, incubated, washed, and reacted with substrate for color change	Laboratory	Moderate, suitable for large-scale screening	Low, equipment and reagents are inexpensive	Simple operation, suitable for large-scale screening, can quantify specific antibodies or antigens	Lower sensitivity, possible cross-reactivity leading to false positives, cannot provide viral load data
PCR	Blood, tissue, saliva, and other liquid samples	Nucleic acid extraction, reverse transcription (if needed), target sequence amplification	Laboratory	High	Moderate, requires high-quality reagents and equipment	High sensitivity, broad applicability, can quantify viral load	Complex sample preparation, long operation time, requires high-quality reagents and skilled operators
Sequencing Technology	Blood, tissue, saliva, and other liquid samples	Requires high-quality DNA/RNA extraction, library preparation, sequencing process is complex	Laboratory	Very high, detects viral mutations at the genome level	High, expensive equipment and data analysis	Provides comprehensive viral genomic information, suitable for mutation and epidemiological studies	High cost, complex data analysis, limited by equipment and technology conditions
IHC	Tissue, cells	Samples need to be fixed, sectioned, stained, and labeled with antibodies	Laboratory	Moderate, suitable for qualitative analysis	Moderate, higher reagent and equipment costs	Can observe the distribution of the virus in tissue, suitable for pathology studies	Time-consuming operation, high background interference, limited sensitivity, cannot provide quantitative analysis
Microarray Technology	Blood, tissue	Samples need to be extracted, labeled, and hybridized with probes on a microarray chip	Laboratory	High, detects multiple genes or viruses simultaneously	High, equipment and chip costs are high	High-throughput, can detect multiple viruses or gene expressions simultaneously	High cost, complex operation, suitable for specific needs in large-scale screening
HTS	Blood, tissue	Samples need to be extracted, sequenced, and analyzed with high-throughput sequencing methods	Laboratory	Very high, provides comprehensive viral genomic information	High, expensive equipment and data analysis	Can obtain comprehensive viral information and mutation data, suitable for studying new viral variants	High cost, complex data processing, limited application range, time-consuming
Isothermal Amplification	Blood, saliva, urine, and other liquid samples	Nucleic acid extraction, isothermal amplification reaction	Laboratory or field environment	Moderate to high, depends on optimization	Moderate, equipment is inexpensive and reagents are expensive	Simple operation, rapid, suitable for on-site rapid detection, applicable in resource-limited areas	Amplification products may be affected by contamination or interference, limited applicability
CRISPR-Cas Technology	Blood, tissue	Nucleic acid extraction, CRISPR-Cas system for editing or detection reaction	Laboratory	High, precise and specific	Moderate, equipment is inexpensive and reagents are expensive	High sensitivity, precise detection, potential for future use in viral detection	Technology is not yet mature, complex operation, requires high-quality reagents and equipment
AI	High-throughput data, imaging data	Requires data collection and processing before applying AI algorithms for analysis	Laboratory environment, computational platform	High, depends on data quality and algorithm optimization	Low, data analysis cost is low	Provides automated data analysis and prediction, suitable for large-scale data processing	Dependent on data quality, may struggle with unknown variants or non-standard data
POCT	Saliva, blood, throat swabs, and other liquid samples	Sample collection, then direct detection without complex processing	On-site detection, grassroots medical environments	Moderate, typically used for preliminary screening	Low, relatively inexpensive equipment, easy to operate	Suitable for on-site rapid detection, simple operation, suitable for large-scale screening	Lower sensitivity and specificity, suitable for screening rather than diagnosis, results may be affected by sample quality
Multi-Omics Analysis	Blood, tissue, cells, and various samples	Requires multi-omics data collection and processing (e.g., genomics, transcriptomics, proteomics)	Laboratory, high-throughput equipment	High, provides comprehensive viral and host response information	High, expensive equipment and analysis	Provides comprehensive viral and host interaction information, suitable for in-depth study of viral mechanisms	Complex data processing, high cost, relies on large sample sizes and multidisciplinary team support

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
