# Peer review of "Advances in Epstein–Barr Virus Detection: From Traditional Methods to Modern Technologies"

_viruses, 2025, doi:10.3390/v17081026_

Round 1

Reviewer 1 Report

Comments and Suggestions for Authors

In this review, the authors systematically summarize the historical development of EBV diagnostic technologies, highlighting the key milestones and future trends in precision medicine and global health readiness. My comments for the review are listed as follows:

  1. In Section 2.3, substantial space has been dedicated to presenting the clinical features of specific diseases associated with EBV infection. It is recommended to add some descriptions of the EBV infection-associated malignancies, such as B lymphocyte-derived lymphoma and epithelium-originated carcinomas like NPC, and the difference in the pattern of EBV infection between these cancers.
  2. In Section 3. Diagnostic Methods, Table 2 simply lists the comparison of different testing methods for EBV. It is suggested to further illustrate the integrated detection of EBV using multiple technologies in applicable scenarios. A diagnostic flowchart is preferable.
  3. The title of Figure 2 does not match the content, it is more likely to be “The diseases induced by EBV infection” rather than “Clinical Syndromes”. In addition, the characters in the circles are a little bit hard to read.

Reviewer 2 Report

Comments and Suggestions for Authors

EBV is a ubiquitous human pathogen that is characterized by a dual replication cycle which enables the virus to both persist for the life of the host and spread to uninfected individuals. Sun et al. has provided a historical and current perspective on the diagnostic detection methods for EBV. This review is a great foundational piece for researchers, clinicians, clinical laboratory personal, and those new to the field. Comments are provided to aid in improving the clarity as well as editorial suggestions.

Specific comments:

-Figure 1: See comments below.

Need to refer to this figure in the main text as “(Figure 1)”.

The color shading of the “Lytic Cycle Inducers” BZLF1 and BRLF1 is very similar to the color used with the EBV virion.

Would suggests providing further details within the main body of the text (2-3 sentences) describing Latency I, II, and III. A reader that is unfamiliar with EBV biology will not know this important designation of these latency phases.

As shown in the figure, will need to also describe the tethering of the circular EBV genome to chromosomes within the main text. A reader unfamiliar with EBV would not understand this diagram.

Within the figure legend would suggests changing the word “invasion” to “infection”.

-Figure 4: See comments below.

Need to refer to this figure in the main text as “(Figure 4)”.

Under the “Application” area of the figure there is a line with “…”; was this intentional or is text missing?

-Table 2: The summary in Table 2 is not in agreement with the statement “…PCR, ELISA, and IHC are commonly utilized but are constrained by limitations in sensitivity, speed, and cost” (Page 17, Paragraph 1). Specifically, within the table the PCR assay is listed as “high” sensitivity while in the text PCR is referred to as “…constrained by limitations in…sensitivity”. Also, ELISA within the table is concluded as a “low” cost but the text concludes ELISA is “…constrained by limitations in…cost”.

-Table 2: The messaging is inconsistent with the information provided for the IF assay. Under advantages the IF assay is “suitable for qualitative and quantitative detection of specific antigens.” However, as a disadvantage it was concluded the IF assay “cannot provide quantitative data.” Based on the scientific literature, one can collect quantitative data from IF experiments. Our lab collects quantitative data from IF experiments and performs statistically analysis with scientific software packages Imaris and Huygens.

-Page 9-10: Within the “3.2.2. High-throughput sequencing” section may want to briefly discuss the EBV studies using single-cell sequencing.

Editorial comments:

-Need to pay particular attention to grammatical errors, sentence structure, spelling, word choice, punctuation, and the use of spaces after a reference.

-Page 2, Paragraph 2: There are “9” human herpesviruses, not “8”. Classification has been updated with HHV-6A and HHV-6B.

-References are missing in the following areas. See below.

Page 1, Paragraph 1.

Page 2, Paragraph 3: “The prevalence of EBV infection is exceptionally high, affecting over 95% of the adult population worldwide”.

Page 4, Paragraph 3.

Page 12, Paragraph 1: “Since the first successful use of CRISPR-Cas9 for gene editing in mouse and human cells in 2013…”

Page 18, Paragraph 3: “faster, more accurate, and more scenario-based”.

Page 15, Paragraph 4: “COVID-19 test to promote the standardization of platforms, and then to expand to other pathogens”.

-Figure 3: Need to refer to this figure in the main text as “(Figure 3)”.

-Page 6, Paragraph 5: Would suggest using the last or full name of “Werner Hele” rather than just their first name.

-Page 9, Paragraph 2: The last two sentences are basically the same. Consider combining.

-Page 9, Paragraph 6: The “birdshot” sequencing technique mentioned in the text is traditional referred to as “shotgun” sequencing. The primary reference (Zeng, J Virol, 2005) also refers to this technique as “shotgun” sequencing.

-Page 10, Paragraph 3: Edit the sentence structure of “…single long sequence sequenced…”.

-Page 10, Paragraph 5: “…utilized to assess the expression of latent versus cleaved genes…”. Based on the provided reference (Lui 2013, Int J Mol Sci), would change the word “cleaved” to “lytic”.

-There are two examples (see below) of incorrect dates being referenced in the text. Authors need to meticulously evaluate all dates that are quoted in the text and confirm with the manuscripts that are referenced.

Page 12, Paragraph 2: “In 2017, and CRISPER-Cas12a-based…”. Based on the provided reference (Chen 2018, Science) the date should be changed from “2017” to “2018”.

Page 12, Paragraph 3: “CRISPR-based Cas13-based viral detection technology, SHERLOCK, was also introduced in 2017.” Based on the provided reference (Myhrvold 2018, Science) the date should be changed from “2017” to “2018”.

-Page 19, Paragraph 1: Define “ctEBV DNA”. Also, it appears that it was misspelled in the text as “cfEBV”.

Comments on the Quality of English Language

The authors need to pay particular attention to grammatical errors, sentence structure, spelling, word choice, punctuation, and the use of spaces after a reference.

Reviewer 3 Report

Comments and Suggestions for Authors

Major Comments:

  1. Lack of clinical relevance and diagnostic clarity:
    The most critical shortcoming of this review is the absence of a clear explanation of the Epstein-Barr virus (EBV) diagnostic methods currently employed in clinical practice, as well as their clinical significance. Currently, there are three major approaches for EBV detection, each with distinct purposes (e.g., assessing infection status, monitoring viral load, or differentiating primary from past infection). However, the authors do not clearly describe these distinctions. Furthermore, the manuscript fails to adequately differentiate between qualitative and quantitative assays, which have fundamentally different implications. This lack of clarity undermines the utility of the review for clinicians and researchers alike.
  2. Insufficient virological perspective:
    The review lacks essential context from a virological standpoint. For instance, although over 90% of adults are EBV-positive, most remain asymptomatic. This point is critical when discussing the clinical significance of diagnostic testing. Unlike acute viral infections such as influenza or SARS-CoV-2, EBV positivity is often of limited diagnostic value unless specific diseases are suspected. Any new diagnostic method should therefore be discussed in terms of its applicability to defined disease contexts and its advantages over existing techniques.
  3. Overall structure and consistency issues:
    The review is missing numerous critical elements and requires substantial revision. In particular:
    • Inconsistent abbreviation usage:
      The manuscript inconsistently introduces abbreviations, often reintroducing full terms after abbreviations have already been defined (e.g., EBV, HHV, EM, IF, VCA, EA, HTS, NPC, IM, qPCR). In other instances, abbreviations are used without ever being defined (e.g., COVID-19, LMP, BZLF1, BRLF1, BMRF1, BALF1, BHRF1, CRISPR). Moreover, some abbreviations appear before their full terms are introduced (e.g., ISH, ELISA, PCR, IHC, EBNA, VCA, EA). This significantly hampers readability and should be standardized throughout the text.
    • Formatting inconsistencies and terminology confusion:
      Terms such as Burkitt lymphoma are unnecessarily bolded without justification. Figure 1 lacks critical components, such as EBER and BART miRNA annotations. Minor typographical errors (e.g., missing spaces: "1980s.In 1990") and inconsistent terminology (e.g., HTS vs. NGS) further detract from the manuscript’s clarity. These should be corrected and unified.
    • Summary section lacks focus:
      The Summary section should offer a concise overview of the entire review, highlighting both the strengths and limitations of conventional and emerging diagnostic methods. As currently written, it fails to fulfill this role.

Comments on the Quality of English Language

This review requires extensive English editing.

Round 2

Reviewer 1 Report

Comments and Suggestions for Authors

The authors have addressed all of the concerns raised. I have no further comments.

Author Response

Thank you for your feedback on our English writing. We truly appreciate your careful review and the time you invested in pointing out the areas for improvement. Following your suggestions, we have thoroughly revised the entire document to enhance its language accuracy, clarity, and fluency.

Reviewer 3 Report

Comments and Suggestions for Authors

This review article has failed to address many of the major concerns raised in the previous peer review and still contains significant issues. Moreover, multiple fundamental errors in the fields of virology and laboratory diagnostics strongly suggest that the manuscript does not meet the standards of a scientific publication.

  1. Unresolved Issues from the Previous Review:
  2. Omission of EBER and BART miRNA
    EBER is a central marker in current EBV-related diagnostics, and BART miRNA has also been proposed as a potential future diagnostic target by several groups. The continued absence of any description of these elements in Figure 1 represents a critical deficiency.
  3. Redundant use of terminology
    Terms such as “Epstein-Barr nuclear antigen” and “Infectious mononucleosis” are repeated multiple times throughout the manuscript, with inconsistent use of abbreviations.
  4. Errors in notation and lack of proofreading
    Frequent typographical issues such as missing spaces before parentheses and the incorrect spelling of Dr. Werner Henle as “Hele” remain uncorrected. These problems indicate that the manuscript has not undergone appropriate English proofreading.
  5. Errors in Virology:
  6. Incorrect number of HHVs
    There are eight known human herpesviruses (HHVs) that infect humans, not nine.
  7. Gene name formatting
    The term “Bam” should be italicized.
    The full name of BZLF1 is “BamHI Z fragment leftward open reading frame 1.”
  8. Latent infection classification
    PTLD is classified as latency type III. EBVaGC is classified as latency type I or I/II, and does not express LMP1. In contrast, NPC is classified as latency type II and expresses LMP1. Such classifications should be based on authoritative virology references such as Fields Virology.
  9. Lack of description of infected cell types and latency patterns in CAEBV
    The manuscript does not specify which cell types are infected in CAEBV or which latency type they belong to.
  10. Misinterpretation of EBV-associated autoimmune diseases
    While there is strong epidemiological support for an association between EBV and multiple sclerosis (MS), the data supporting links to systemic lupus erythematosus (SLE) and rheumatoid arthritis are more limited. In particular, rheumatoid arthritis is not discussed in Fields Virology.
  11. Issues in Laboratory Diagnostics:
  1. Lack of clear classification of clinically used diagnostic methods
    In clinical practice, EBV infection is generally diagnosed using three main approaches: ELISA for antibody titers, ISH for tissue-based diagnosis, and qPCR for measuring viral DNA levels in blood. Other techniques such as IHC, multiplex PCR, and FACS-ISH are used more rarely and are mostly limited to research settings. Despite this being pointed out in the previous review, the current version still lacks clear classification of these methods by usage.
  2. Inaccurate statements regarding POCT
    Descriptions of POCT (point-of-care testing) are vague. The cited references suggest that EBV POCT technologies are still under development and may not be in clinical use, even in countries such as China. The authors should clearly state where such tests are used and whether they are approved for clinical application.
  3. Inappropriate citation of references
    Reference 119 describes a modified method for qPCR and is unrelated to POCT. This reflects a lack of precision in selecting and interpreting relevant literature.

Comments on the Quality of English Language

There are many problems with the English expression. A large-scale English proofreading by the editorial department will be necessary.

Author Response

We would like to express our sincere gratitude for your insightful comments and suggestions on our manuscript. Based on your feedback, we have carefully revised the paper to address the concerns raised. Please find below a summary of the key modifications, and we appreciate your further review and guidance.

Round 3

Reviewer 3 Report

Comments and Suggestions for Authors

Unfortunately, even at this stage, the submitted review still contains numerous errors. The quality of the English editing appears to be very low; therefore, I strongly recommend using the publisher’s English editing service.

  1. BART miRNAs are also expressed during type III latency. However, this is not reflected in Figure 1.
  2. The terms “LMP1” and “LMP-1” are used inconsistently.
  3. “Viral capsid antigen” is inconsistently abbreviated as both “VCA” and “VAC.” Multiple instances of the full term are also used.
  4. The full term “enzyme-linked immunosorbent assay” appears before Section 3.1.1, but an abbreviation should follow.
  5. “PCR” appears before Section 3.1.3, but the full term is not provided until 3.1.5.
  6. Based on the usage elsewhere, “Henle” likely refers to Warner Henle.
  7. “dPCR” likely refers to droplet digital PCR (ddPCR).
  8. “MDCA” appears in its abbreviated form without being defined first.

Comments on the Quality of English Language

Due to the large number of errors, I recommend that the publisher provide thorough and professional English editing assistance.

Author Response

Thank you very much for your valuable advice. We have revised the manuscript with the assistance of MDPI's English Editing Services, and we hope the revisions meet your expectations.
